# Mechanical Forces and Their Effect on the Ribosome and Protein Translation Machinery

**DOI:** 10.3390/cells9030650

**Published:** 2020-03-07

**Authors:** Lisa J. Simpson, Ellie Tzima, John S. Reader

**Affiliations:** Wellcome Centre for Human Genetics, Division of Cardiovascular Medicine, Radcliffe Department of Medicine, University of Oxford, Oxford. OX3 7BN, UK; lisa.simpson@worc.ox.ac.uk (L.J.S.); ellie@well.ox.ac.uk (E.T.)

**Keywords:** forces, ribosome, protein translation

## Abstract

Mechanical forces acting on biological systems, at both the macroscopic and microscopic levels, play an important part in shaping cellular phenotypes. There is a growing realization that biomolecules that respond to force directly applied to them, or via mechano-sensitive signalling pathways, can produce profound changes to not only transcriptional pathways, but also in protein translation. Forces naturally occurring at the molecular level can impact the rate at which the bacterial ribosome translates messenger RNA (mRNA) transcripts and influence processes such as co-translational folding of a nascent protein as it exits the ribosome. In eukaryotes, force can also be transduced at the cellular level by the cytoskeleton, the cell’s internal filamentous network. The cytoskeleton closely associates with components of the translational machinery such as ribosomes and elongation factors and, as such, is a crucial determinant of localized protein translation. In this review we will give (1) a brief overview of protein translation in bacteria and eukaryotes and then discuss (2) how mechanical forces are directly involved with ribosomes during active protein synthesis and (3) how eukaryotic ribosomes and other protein translation machinery intimately associates with the mechanosensitive cytoskeleton network.

## 1. Protein Translation in Bacteria and Eukaryotes

Protein synthesis represents a fundamental process allowing for cellular control of gene expression in conjunction with mRNA transcription. Translational regulation allows cells to produce proteins in a rapid and localized manner in response to stress and growth signals [1]. It also maximizes the efficiency of a process that requires significant cellular resources to produce massive protein-making machines, the ribosomes (> 2.6 MDa) and the large retinue of ancillary proteins and tRNAs required for protein synthesis. The ancient role of protein synthesis in life’s history means that many functional aspects and components of this machinery are highly conserved between all three domains of life [2,3]. As a consequence, protein translation in bacteria and eukaryotes can be divided into the same stages: initiation, elongation and termination [4]. Although the initiation and termination phases have noticeable differences, the core elongation phase of translation are very similar.

In bacteria, the ribosomes are composed of two ribonucleoprotein particles: the large (50S) and small (30S) subunits. The small 30S ‘decoding’ subunit binds to the mRNA transcript and is the site where an in-frame codon on the mRNA transcript is interrogated by the bases within the anticodon loop of an aminoacyl-tRNA respectively in order to produce a correct match. The large 50S subunit contains the peptidyl transferase centre, the catalytic ribosomal RNA structure responsible for peptide bond formation [5]. The initiation phase in *Escherichia coli* is triggered when the AUG start codon downstream of the Shine-Dalgarno (SD) sequence is located by the GTP-bound ribosomal 30S subunit in complex with IF1, IF3 and an IF2•GTP•N-formyl-methionyl-initiator tRNA (fMet-tRNA_i_^Met^) complex, with the tRNA_i_^Met^ anticodon stem loop (ASL) bound to the peptidyl or P-site of the 30S subunit (Figure 1A). The SD sequence is complementary to the anti-SD sequence found in the 16S rRNA and the base-pairing interaction between the two helps to align the ribosomes in the correct position for initiation [6].

The elongation stage commences when an aminoacyl-tRNA (aa-tRNA), bound to a GTP-activated elongation factor-thermo unstable protein (EF-Tu•GTP), uses its ASL to interrogate the second codon position in the aminoacyl or A -site of the ribosome. If a successful codon–anticodon pairing occurs, conformational changes occur in the ribosome leading to hydrolysis of the EF-Tu-bound GTP to GDP and subsequent release of the A-site-bound aa-tRNA, whose CCA 3′-end then moves up into the peptidyl transferase centre. Peptide bond formation is then catalysed between the carboxyester group of the fMet-tRNA_i_^Met^ and the attacking N-terminal amine of the amino acid ligated to the incoming aa-tRNA in the A-site. In this manner, the growing peptide chain relocates to the A-site tRNA. The deacylated peptidyl-tRNA then leaves the P-site to relocate to a third exit or E site [7]. In a coordinated conformational change, a second elongation factor G (EF-G), which structurally mimics an aa-tRNA-bound-EF-Tu [8], then associates with the ribosome, and in a GTP-dependent, ratchet-like mechanism moves the A-site peptidyl-tRNA into the P site location in a process called mRNA–tRNA translocation (see more details in Section 2.1). This process is repeated for the next incoming EF-Tu•GTP•aa-tRNA corresponding to the third codon, and so on, along the mRNA reading frame, while the growing nascent chain extends through a polypeptide exit tunnel until it protrudes outside of the back of the large subunit. Finally, termination occurs when a stop codon is encountered (UAA, UAG or UGA) and recognized by termination release factors RF1 and RF2 [9], which excise the folding nascent chain, as it concomitantly undergoes quality control by protein chaperones before it becomes a fully functional protein [10].

In eukaryotes, ribosomes are greater in size but are still composed of a large (60S) and a small (40S) subunit, each of which carries out the same primary functions as their bacterial counterparts. A second important difference is that transcription and translation are completely segregated processes in eukaryotes, whereas bacteria can immediately translate mRNA as it is being transcribed (co-transcription) by the RNA polymerase [9]. Bacterial ribosomes can bind mRNA transcripts immediately after their release from an RNA polymerase to enable rapid translation of proteins, independent of quality control checks. Conversely, mRNA is first transcribed in the eukaryotic nucleus and requires export into the cytoplasm to allow association with the correct complex of initiation factors and the small 40S subunit before formation of complete ribosomes and protein synthesis can commence.

The first phase of initiation is governed a collection of eukaryotic translation initiation factors (eIFs), which is a rate-limiting phase, as mRNA transcripts must compete for mature ribosomal subunits and eIFs [11]. In eukaryotes, there are more quality control checks on stable mRNAs before translation initiation, including 5′–7-methylguanosine cap recognition and mechanisms to monitor the length of the poly-A tail. In eukaryotic cells undergoing cap-dependent translation, a pre-formed 43S pre-initiation complex (PIC) forms from the small (40S) ribosomal subunit, a Met-tRNA_i_^Met^ bound to a GTP-activated eIF2, and the other factors eIF1, 1A, 3 and 5. The PIC then binds to a 5′-capped mRNA transcript via a cap recognition complex formed from eIF4E, G, A and F [4] and then actively scans along the 5′-untranslated region (5′-UTR) until locating for the AUG start codon. This leads to hydrolysis of the GTP bound to Met-tRNA_i_^Met^•eIF2 catalysed by the eIF5 GTPase, releasing eIF2•GDP from the Met-tRNA_i_^Met^ , allowing recruitment of the 60S large ribosomal subunit and entry into the elongation stage. Elongation factors eEF1A and eEF2 are eukaryotic homologs of the bacterial elongation factors EF-Tu and EF-G and conserve the same core functions in translation. Interestingly, eEF1A has additional functions in eukaryotic cells, involving the mechanically active actin cytoskeleton and which are detailed in Section 3. Finally, termination occurs when the highly conserved stop codons UAA, UAG, or UGA enter the decoding centre of ribosome and are recognized by a single release factor eRF1 (functionally equivalent to bacterial RF1 and RF2) in eRF1-eRF3-GTP complex stimulating release of the nascent polypeptide chain from the ribosome [12].

A level of translational regulation independent of transcription is beneficial for the cell where a more rapid and dynamic fine-tuning of cellular phenotype is required in times of cellular stress, such as nutrient deprivation or hypoxia [4]. Modulation of eukaryotic translation typically occurs at the initiation stage through a variety of signalling pathways such as Ras-ERK and PI3K-mTOR [13] which modify the formation of the mRNA 5′-cap recognition complex [4]. A second key regulatory mechanism is the phosphorylation of the α subunit of eIF2 leading to downregulation of protein initiation by a family of eIF2α kinases which are activated by four molecular stresses: (1) GCN2-tRNA-dependent amino acid deprivation and UV radiation, (2) PKR-ds RNA and viral infections, (3) PERK- ER stress and hypoxia, and (4) HRI-heme levels and heavy metals [14]. Although the majority of translational control is observed at the level of initiation, the elongation rate of a nascent polypeptide chain can be subject to ribosomal stalling, also referred to as ‘translational pauses’. Stalls can be transient and resolved quickly or they can bring elongation to an abrupt, irreversible halt. A recent ribosomal profiling meta-analysis revealed that ribosomes do indeed pause at sites where codons are rare or non-optimal [15]. Faster translation of optimal codons or translational pauses at rare codons could be a natural feature of the system to fine-tune the translational rate of these proteins in biology, and also allow for correct folding of on-pathway intermediate states as the nascent polypeptide chain protrudes from the ribosomal exit tunnel on the large 60S subunit in a process known as co-translational folding (see Section 2.3). In addition, regulation of the aminoacyl-tRNA synthetase activity, the enzymes responsible for ligating an amino acid on to its canonical tRNA isoacceptor, can lead to differential levels of each charged tRNA and, therefore, influence the translation rate.

## 2. Forces Generated at the Level of the Ribosomal Machinery

### 2.1. EF-G and the Ribosome Acts as a Molecular Ratchet System to Drive mRNA–tRNA Translocation

In the early 2000s, X-ray structures were solved for individual ribosomal subunits [5,16,17,18]. Concomitant with these studies were improvements in cryo-electron microscopy (cryo-EM), which captured detailed snapshots of the conformations populated by the interacting ribosomes, elongation factors and guanine nucleotide analogs, and acylated-tRNAs during the translation elongation cycle [19]. These advances increased our understanding of the conformational changes exhibited by the ribosome during mRNA–tRNA translocation. It was proposed that the elongation factor EF-G is a driving force in this process, as it was reported to act as a ‘motor’ protein that couples the hydrolysis of GTP to mRNA/tRNA translocation in the ribosome [20]. 3D cryo-EM models revealed a ~6° rotation of the 30S subunit relative to the 50S subunit when EF-G•GTP was bound to the ribosome [21]. The 30S subunit rotated back when bound to EF-G•GDP and fusidic acid, an antibiotic that prevents turnover of EF-G when bound to the ribosome. There were also significant changes in the conformation of the ribosome subunits upon rotation, including opening of the mRNA channel when EF-G•GTP was bound. These results suggested that the GTP/GDP-dependent conformational states of EF-G and the rotating subunits of the ribosome acted together in a ‘ratchet-like’ mechanism whose movements were intimately involved in translocation of the mRNA and tRNAs [21] (Figure 1A). 

Later studies employed single molecule Förster Resonance Energy Transfer (smFRET) to monitor the rotational movements of 50S and 30S bacterial ribosomal subunits, each labelled with a fluorophore and on an immobilized mRNA strand [22,23]. These experiments revealed that the ribosomal subunits rotate spontaneously and reversibly due to Brownian motion. During this process, the 3’ ends of the tRNAs present in the A and P site of the 50S subunit move into a hybrid A/P and P/E state. In a subsequent irreversible step, EF-G facilitates translocation of the mRNA and the anticodon ASL from the A and P site to the P and E site of the 30S. Using single-molecule polarized fluorescence microscopy, Chen et al. also demonstrated that EF-G uses a power-stroke and Brownian- mechanism to achieve translocation of mRNA-tRNA [24].

To obtain detailed insights into the translocation of mRNA/tRNA, it has proved to be important to directly measure the forces produced by an actively translating ribosome. This seemingly experimentally difficult feat has been achieved in a number of studies by utilizing optical tweezers, which use a focused laser beam to trap and manipulate microscopic objects [25,26]. This is discussed in detail in the section below.

### 2.2. mRNA Secondary Structures as Mechanical Barriers to the Ribosome

Strands of mRNA can adopt different structures and shapes at different regions along their sequence such as hairpins and pseudoknots [27]. These structures need to be unfolded before translation can proceed and force is required for this event. Liu et al. [28] found that the maximum force generated by the ribosome (around 13pN) was just sufficient to untangle these mRNA structures, highlighting that their presence can influence translation rates (Figure 1A). It is therefore important to note that force is not just a natural by-product of chemical reactions occurring during translation, but that in its own right helps regulate translation, as the ribosome has to overcome the mechanical barrier imposed by mRNA internal structures. Interestingly, the force generated by the ribosome to unfold temporary mRNA structures is comparable to the force required for the ribosome to overcome stalling along the mRNA transcript [29]. Ribosome pausing, whether through stalling or the presence of secondary mRNA structures, will not only regulate translation rates but also processes such as co-translational folding (see Section 2.3) and frameshifting [30].

Similar to the above observations, Qu et al. also demonstrated that application of force at the ends of an mRNA hairpin, thereby unwinding its secondary structure, results in increased translation rates [31]. Following on from this work, a recent study by Desai and colleagues [32] described a ‘gear shift’ mechanism that ribosomes adopt in order to overcome the mechanical barrier of large mRNA hairpins they encounter. They used a method combining single-molecule high-resolution optical tweezers with fluorescence measurements to directly measure hairpin unwinding whilst simultaneously visualizing EF-G-activated translocation. They found that a hairpin, irrespective of how mechanically stable or strong it was, would always be opened after EF-G binding and therefore demonstrated that EF-G-catalysed translocation and hairpin unwinding are coupled. By using force to modulate the magnitude of the mRNA hairpin barrier, they showed that ribosomes respond to strong barriers by switching to a ‘slow gear’. This is brought about by an allosteric switch in the ribosome itself that they hypothesized allows it to overcome barriers by capitalising on thermal fluctuations [32].

### 2.3. Forces on the Nascent Polypeptide Generated during co-Translation Folding

During the elongation stage of translation, the growing nascent polypeptide chain emerges from the polypeptide exit tunnel, located on the large ribosomal subunit, and starts to fold into discrete secondary structures. It has been shown that the protruding polypeptide chain, once partially folded, generates a ‘tugging’ force on the residual chain that propagates back through the exit tunnel to the peptidyl-tRNA located in the catalytic centre of the ribosome (Figure 1A). These mechanical forces can alter the speed of ribosomal protein synthesis [33,34,35,36] by acting as a feedback to alter the energy barrier to peptide bond formation. This concept is known as co-translational folding, and these pulling forces acting on the nascent chain not only influence translation rates [33,37,38], but also serve as a feedback mechanism on the folding process itself [34,39].

Depending on the protein size, co-translational folding can occur while the nascent protein is still within the ribosome exit tunnel [40] or when it is outside [33]; in both cases, this has been shown to produce pulling forces on the new polypeptide chain. Studies have most commonly used translational arrest peptides (APs), which interact with particular regions of the polypeptide exit tunnel and halt translation [41], as a means to detect forces arising from co-translational folding. The *E.coli* SecM protein AP has been frequently utilized to measure the force generated through protein folding. Goldman et al. demonstrated the folding of the large Top7 protein outside the ribosome exit tunnel produces force sensed through the SecM AP [33]. Nilsson et al. used SecM force detection to reveal that small protein ADR1a folds whilst still in the ribosomal exit tunnel [40]. Using APs, Farías-Rico et al. analysed the co-translational folding of eight protein domains of varying size, fold type, thermodynamic stability and net charge [42]. They found that the pulling force generated on the nascent polypeptide as it folds is proportional to the thermodynamic stability of the folded state. 

Leininger et al. [36] investigated what specific protein domain characteristics dictate the magnitude of the pulling force exerted by folding proteins. Using molecular simulations and statistical mechanical modelling, they reported that domain stability, topology and translation speed determine the magnitude of force generation and point out that forces measured on arrested ribosomes are different from those measured on translating ribosomes. Other studies have found that even before a nascent chain folds, the unfolded state can produce an entropic pulling force on the ribosome [35], and as the chain protrudes from the ribosome it interacts with the outer surface of the ribosome, which has a negative electrostatic potential, giving rise to electrostatic attractions and forces which direct local folding dynamics [43]. The outer ribosomal surface is known to constrain the mobility of polypeptide domains as they emerge from the exit tunnel, slowing down the polypeptide folding–unfolding transitions [44].

Most studies have evaluated pulling forces during co-translational folding in isolated in vitro systems without other factors which assist protein dynamics and folding outside the ribosomal exit tunnel such as chaperones. This has been due to experimental limitations in measuring and monitoring force-driven-single protein folding. It has therefore been difficult to fully understand the direct influence of chaperones on protein folding under force. Nilsson et al. elucidated that the presence of chaperones did not affect the folding forces generated by smaller proteins inside the ribosome exit tunnel using force-sensing AP SecM [34]. However, larger proteins folding outside the exit tunnel were influenced by the presence of the chaperone, Trigger Factor (TF). TF appeared to influence the force profile by generally reducing the force exerted on the new protein and slowing the final folding confirmation [34]. The TF chaperone associates with the ribosome and is poised to interact with newly synthesized polypeptides as they emerge from the ribosome [45]. Kaiser et al. used fluorescence spectroscopy to observe in real-time the actions of TF on translating ribosomes and found that it interacts with both the ribosome and the nascent polypeptide to prevent protein misfolding and aggregation by preventing hydrophobic regions in the unfolded nascent chain interacting [46].To further elucidate the relationship between force, the nascent polypeptide folding on leaving the ribosome and cytoplasmic chaperones like TF, Haldar et al. used force spectroscopy experiments to record protein folding events in real time in the presence of TF [47]. The presence of TF increased the probability of protein folding against force and increased the refolding speed of protein L. The folding reaction was catalysed by TF in a force-dependent manner, and as the force applied was increased, more TF was required to rescue the folding process.

Chaperones play an even more prominent role in the assembly of large multi-domain proteins, which are more prone to inter-domain misfolding events [48,49,50]. A recent study by Liu et al. used optical tweezers to better to evaluate the synthesis and early folding intermediates of the multi-domain protein, EF-G, an essential mediator of ribosome mRNA translocation (as detailed in Section 1 and 2a) [49]. This study demonstrated that both the ribosome and TF help reduce any inter-domain misfolding so that the N-terminal G domain can fold efficiently. They observe that early and proper folding of the N-terminal domain is a necessary step to ensure ordered subsequent folding of other domains. Notably, they also observed new sections of polypeptide emerging from the ribosome exit tunnel can still interact with, and denature, domains that have already folded. They demonstrate that the chaperone TF helps to protect against denaturation and therefore augments multi-domain folding.

## 3. Cell Mechanics and the Protein Translation Machinery

The cytoskeleton not only provides structural support, but also determines the mechanical properties of cells and acts a signalling platform that that regulates the activity and subcellular localization of proteins and organelles. The cytoskeleton is composed of actin filaments, tubulin-based microtubules and intermediate filaments. Cryo-EM studies have highlighted the close interaction between the actin cytoskeleton and components of the protein translational machinery, including, but not limited to, ribosomes [51]. Indeed, mRNA transcripts, polysomes, eukaryotic initiation and elongation factors and aminoacyl-tRNA synthetases have all been shown to associate with the cytoskeleton [52]. This proximity is functionally important because efficient protein translation depends on an intact cytoskeleton, and deletion of cytoskeleton-regulating proteins can affect the initiation of protein translation [53,54]. Additionally, the cytoskeleton is crucial for the transport of specific mRNAs as well as their spatially localized translation in several organisms [55]. It is therefore appreciated that there is reciprocal regulation between the processes of protein translation and cytoskeleton remodelling (Figure 1B).

Pioneering studies performed in the 1970s reported the close physical association between components of the cytoskeleton and ribosomes [56,57,58], while cryoelectron tomography revealed the three-dimensional organization of the filamentous cytoskeleton linked to polysomes [59]. Ribosomes have also been shown to associate with microtubules and intermediate filaments, and these associations might be cell type and context specific [60,61]. In addition to the cytoskeleton, components of the translation machinery have also been observed in other mechanically active areas within the cell, including focal adhesions. Using scanning electron microscopy, ribosomes were first observed at the leading edge of migrating fibroblasts by [62]. Since then, further work has elucidated the recruitment and movement of ribosomes, mRNA and other protein translation factors to focal adhesion complexes and at cell protrusions during migration [63,64,65,66,67,68,69]. Using immunofluorescence, Willett et al. [68] showed that small ribosomal subunits (40S) marked by rpS3 staining, co-localized with β3 integrin-enriched adhesion complexes in fibroblasts migrating over a mixed extracellular matrix composition. Further, they showed that the eukaryotic initiation factor 4E (eIF4E) co-stained with the focal adhesion protein talin along the leading edge of migrating cells [68] (Figure 1B).

Perhaps the most studied cytoskeleton-associated component of the protein translation machinery is one of the most abundant protein synthesis factors, eukaryotic translation elongation factor 1A (eEF1A) (Figure 1B). The canonical role of eEF1A is to bind and deliver cognate aa-tRNA to the A site of the ribosome during translation elongation; once a codon/anticodon match is detected, eEF1A deposits the aa-tRNA and is released from the ribosome. eEF1A was identified as an actin-binding protein [70] that can both bind and bundle actin filaments in vitro [71] importantly, binding of aa-tRNA and actin to eEF1A is mutually exclusive [72]. Through its close interaction, eEF1A plays an important role in mediating cytoskeletal organisation as mutations in the actin binding ability of eE1FA leads to disruption of the actin cytoskeleton [54]. Because eEF1A was shown to cross-link actin filaments via a unique bonding rule [73], it was proposed that the resulting actin structure could serve as a scaffold for mRNA. Indeed, Liu et al. demonstrated that eEF1A co-localizes with β-actin mRNA and actin filaments in protrusions in vivo and binds directly to F-actin and β-actin mRNA simultaneously in vitro [64]. Disruption of the actin binding ability of eEF1A leads to reduced interaction with and, therefore, mis-localization of β-actin mRNA within migrating fibroblasts [64]. Another study that highlighted the importance of translation initiation factors in controlling cytoskeletal dynamics is that of Fujimura et al. [74]. They used proteomic profiling and bioinformatics analyses to show that the eukaryotic initiation factor 5A (eIF5A) is a master regulator of an integrated network of cytoskeleton-regulatory proteins involved in cell migration. Further investigation into this network highlighted that eIF5A mediates tumour cell movement and metastasis through modulation of RhoA/ROCK protein expression levels.

Evidence for the role of the cytoskeleton in regulating protein synthesis is increasing. In mammalian cells, depolymerization of actin filaments leads to a major reduction in resumption of protein translation in response to cold shock [53], whereas in yeast an intact actin filament network is a pre-requisite for regulation of protein synthesis [54]. More recently, it was demonstrated that disruption of filamentous actin in mammalian cells impedes translation via phosphorylation of the α-subunit of eukaryotic initiation factor eIF2, the heterotrimeric factor that delivers the initiator methionyl-tRNA^Met^ to the ribosome [52]. A wide range of stressors, including amino acid starvation, leads to phosphorylation of eIF2α on Ser51 and downstream inhibition of general protein synthesis [75]. One of the kinases that phosphorylates eIF2α is GCN2 (or eIF2AK4), which is activated in response to low levels of amino acid, serum or glucose and also by proteasome inhibition [76]. Under normal amino acid conditions, GCN2 binds to eEF1A, and this complex inhibits GCN2-dependent phosphorylation of eIF2α; uncharged tRNA displaces eEF1A from GCN2, thus allowing GCN2 to phosphorylate eIF2α. Importantly, mutations in eEF1A that affect aminoacyl-tRNA binding simultaneously cause actin binding defects and increased phosphorylation of GCN2-dependent eIF2α phosphorylation [77]. Silva et al. [52] showed that an increased G-actin: F-actin ratio promotes the displacement of eEF1A from GCN2 and accumulation of deacylated tRNAs, both of which contribute to the activation of GCN2, phosphorylation of eIF2α and attenuation of global translation.

The cytoskeleton not only acts as a conduit for force throughout the cell (mechanotransduction), but also as a transporter of mRNA transcripts and translational factors to areas of the cell that require local protein synthesis, e.g., neuronal growth cones, focal adhesion assembly (Figure 1B). Singer’s group visualized specific mRNAs on the filamentous cytoskeleton using electron microscopy and in situ hybridization [78], and early evidence for localized hotspots of translation was discovered in the 1980s by Steward and Levy who observed polysomes at synaptic sites in neurons [79]. It is now well appreciated that specific proteins need to be translated locally instead of being transported to the site where they are needed and that local protein synthesis is crucial in specialized neuronal compartments, such as growth cones, axons and dendritic spines [79,80]. In a recent study by Hafner and colleagues, RNA sequencing and super resolution microscopy were utilized to highlight a plethora of mRNAs localized in axon terminals along with the translational machinery required for protein [81]. They also demonstrated that proteins required for synaptic transmission were synthesized within the axon terminal, indicating local protein supply can meet demand as a result of regional translation. Local translation requires an intact cytoskeleton and is critical for neuronal development and synaptic function, including the formation and storage of long-term memory [82]. In addition to the more specialized neuron compartments, localized mRNAs have been shown to spatially regulate translation in the context of cell migration and focal adhesion remodeling [83] Real time in vivo fluorescent tracking of the coupled behaviour of ribosomes and β-actin mRNA at the cell’s leading edge showed that spatial translation of β-actin mRNA preferentially occurs at the leading edge of migrating fibroblasts, around vinculin-positive focal adhesions [84]. Like β-actin mRNA, the mRNA encoding proteins involved in actin dynamics, filamin and α-actinin are also locally recruited to polysomes during cell migration [85].

## 4. Concluding Remarks

Increasing experimental evidence posits the importance of mechanobiology in regulating the fundamental cellular process of protein translation. Mechanical forces can affect the rate at which bacterial ribosomes decode mRNA transcripts as well as co-translational folding of nascent proteins as they exit the ribosome. In eukaryotes, the close physical integration between the mechanosensitive cytoskeleton and ribosomal machinery facilitates dynamic bi-directional control. This mechanoreciprocity allows cells to regulate protein translation both globally and locally. Ultimately, mechanobiology offers the potential to bring new insights not only into the fundamentals of protein translation, but also provide the aetiology and prevention of diseases where disorders of protein translation are a central feature, including heart failure, cancer and neurological diseases.

## Figures and Tables

**Figure 1 cells-09-00650-f001:**
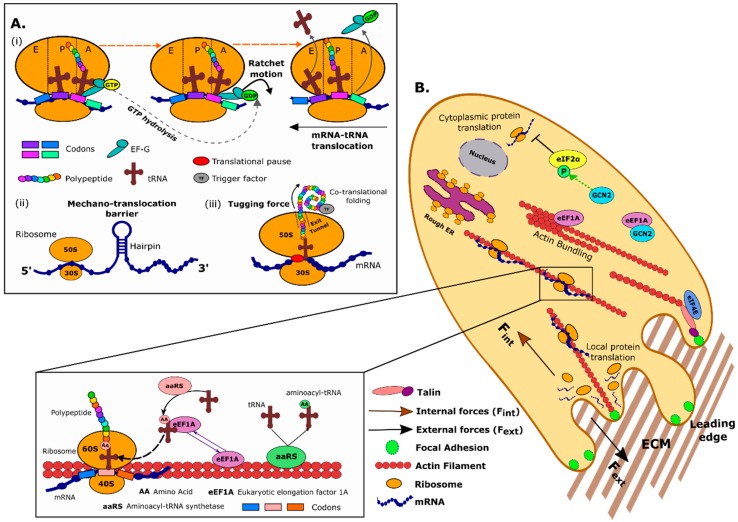
Mechanics in protein translation. (**A**) Schematic of role of forces in prokaryotic protein translation. (i) Forces are generated during the ratchet motion—a molecular motion that moves the tRNA and mRNA so that the next codon in sequence can be read. (ii) Force is required to unwind downstream mRNA hairpins, that act as a mechano-translocation barrier, and allow ribosomes to decode the codon sequence hidden within. (iii) Tugging forces also occur on the polypeptide chain as it folds which are propagated back to the ribosomal machinery; these forces help overcome translational pauses. (**B**) In eukaryotic cells, forces can be generated internally, i.e., by the dynamic cytoskeleton, and/or externally by the ECM via large multiprotein adhesions (focal adhesions). The actin filament network modulates mechanotransduction, and components of the protein translation machinery (e.g., eEF1A and eIF4E as well as mRNAs) associate with actin filaments. Localized protein synthesis allows for fast cell responses to internal and external forces. Abbreviations: AA = amino acid, mRNA = messenger ribonucleic acid, tRNA = transfer ribonucleic acid, TF = trigger factor, EF-G = elongation factor G, GTP = guanosine triphosphate, GDP = guanosine diphosphate, ECM = extracellular matrix, ER = endoplasmic reticulum, eIF2α = eukaryotic initiation factor 2α, eIF4E = eukaryotic initiation factor 4E, eEF1A = eukaryotic elongation factor 1A, GCN2 = general control nonderepressible 2, P = phosphorylation, aaRS = aminoacyl-tRNA synthetase.

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
