# Peer review of "Mechanical Forces and Their Effect on the Ribosome and Protein Translation Machinery"

_cells, 2020, doi:10.3390/cells9030650_

Round 1

Reviewer 1 Report

This is a well-written and very readable review that gives a comprehensive view of a very interesting topic, the mechanical forces acting on translation in diverse physiological contexts. I’d only suggest to add some conclusive remarks to highlight the main take-home message(s) and perhaps to indicate future research directions.

Author Response

We thank this Reviewer for their positive comments and suggestions. We have now included a section at the end of the manuscript on concluding remarks.

Reviewer 2 Report

This is a comprehensive and well-written review on the effects of mechanical forces on ribosome and protein translation machinery. 

General comments.

The authors must add a general conclusion and perspectives. This would help to improve the general interest of the review. Including key questions/issues that remain to be understood in the field would imporove the relevance of the review.

Minor comments. There are a number of typo errors throughout the manuscript:

lines 89-92. please rephrase

line 93: missing word?

line 254:delete "that"

line 273: missing word?

Author Response

We thank the Reviewer for their comments and suggestions. We have now added a conclusion section to the manuscript and have corrected the typos throughout the manuscript.